# Spatial Distribution of Land Surface Temperatures in Kuwait: Urban Heat and Cool Islands

**DOI:** 10.3390/ijerph17092993

**Published:** 2020-04-26

**Authors:** Barrak Alahmad, Linda Powers Tomasso, Ali Al-Hemoud, Peter James, Petros Koutrakis

**Affiliations:** 1Department of Environmental Health, Harvard T.H. Chan School of Public Health, Boston, MA 02115, USA; 2Environmental and Occupational Health Department, Faculty of Public Health, Kuwait University, Kuwait City 24923, Kuwait; 3Environment and Life Sciences Research Center, Kuwait Institute for Scientific Research, Kuwait City 24885, Kuwait; 4Department of Population Medicine, Harvard Medical School and Harvard Pilgrim Health Care Institute, Boston, MA 02115, USA

**Keywords:** urban heat island, urban cool island, Kuwait, land surface temperature, MODIS, Google earth engine, climate change

## Abstract

The global rise of urbanization has led to the formation of surface urban heat islands and surface urban cool islands. Urban heat islands have been shown to increase thermal discomfort, which increases heat stress and heat-related diseases. In Kuwait, a hyper-arid desert climate, most of the population lives in urban and suburban areas. In this study, we characterized the spatial distribution of land surface temperatures and investigated the presence of urban heat and cool effects in Kuwait. We used historical Moderate-Resolution Imaging Spectroradiometer (MODIS) Terra satellite 8-day composite land surface temperature (LST) from 2001 to 2017. We calculated the average LSTs of the urban/suburban governorates and compared them to the average LSTs of the rural and barren lands. We repeated the analysis for daytime and nighttime LST. During the day, the temperature difference (urban/suburban minus versus governorates) was −1.1 °C (95% CI; −1.2, −1.00, *p* < 0.001) indicating a daytime urban cool island. At night, the temperature difference (urban/suburban versus rural governorates) became 3.6 °C (95% CI; 3.5, 3.7, *p* < 0.001) indicating a nighttime urban heat island. In light of rising temperatures in Kuwait, this work can inform climate change adaptation efforts in the country including urban planning policies, but also has the potential to improve temperature exposure assessment for future population health studies.

## 1. Introduction

The rise of urbanization and urban development has been associated with energy-intensive land use that replaces natural land cover. Urban structures, such as buildings, pavements, and asphalt, have direct influence on surface temperatures [1]. The use of air conditioners [2], reduced air flow from narrow streets and tall buildings [3], and the absorbance of solar energy from dark surfaces [4] are all mechanisms that contribute to a well-studied phenomenon of urban heat islands, where urban areas are hotter than surrounding rural areas. Much of the research in previous decades has been dominated by the characterization, health effects, and policy implications of the urban heat island. Urban heat islands have been shown to have adverse health impacts. First, they have been linked to an increase in the frequency and magnitude of thermal discomfort, which increases heat stress and heat-related diseases [5,6,7]. Secondly, the urban warming also enhances the photochemical reaction that leads to higher levels of ozone (O_3_) [8]. Environmental sustainability is also affected by urban warming. It is associated with an increase in electricity consumption [9], per-capita water consumption [10], and extensive irrigation of green cover [11].

Yet, the surface urban cool island is another phenomenon that, contrary to the heat island, occurs when surface temperatures of rural areas are hotter than urban areas [12]. Compared to the urban heat island, the urban cool island is relatively weaker in intensity, always occurs during the day, and is commonly seen in semi-arid and arid regions [13]. The mechanisms and hypotheses behind urban cool islands vary by geographic locations; examples include increased shady areas due to tall buildings [14], inhibition of early morning advection events by warm continental air [15], sea breeze [16], and cooling from evaporation of moist urban soil compared to dry rural soil in arid desert climates [13]. It is unknown whether the daytime urban cool island mitigates the adverse outcomes of heat islands.

While there is growing evidence that temperature affects health outcomes, there is limited literature on the spatial distribution of temperature in Kuwait. Kuwait is a desert country with a hyper-arid climate. Ambient temperatures in Kuwait frequently rise above 50 °C during the long summer that extends for more than five months of the year (from 21 May to 4 November). In 2016, Asia’s highest ambient temperature ever recorded (54.0 °C) was seen in Mitribah, Kuwait [17]. With the current rate of climate change, projections of future temperatures towards the end of this century in the region will possibly exceed the threshold of human adaptability [18]. Our previous studies in Kuwait revealed an alarming increase in the overall mortality risk and cardiovascular mortality risk among vulnerable subpopulations during days of extremely high ambient temperatures measured from monitoring stations across the country [19,20]. Kuwait is also a small urban country; the vast majority of the population lives in urban and suburban neighborhood units around the Kuwait City metropolitan area [21]. Spatial characterization of the distribution of temperature exposure can be critical for population health studies.

Land surface temperature (LST) represents the radiative temperature of any land surface, such as soil, grass, pavements, asphalt, or roofs of buildings [22]. It can, therefore, be directly affected by albedo, vegetation cover, and soil moisture. On the other hand, conventional ambient temperatures measure the temperature of the air near the surface. Land surface temperatures are usually measured by remote sensing techniques that retrieve satellite thermal infrared data [22], while ambient temperatures are measured by ground thermometers. Most temperature-related health studies rely on ambient temperatures to reflect the human microenvironmental exposure in their exposure-response estimation [23,24]. The problem of using ambient air temperature data to investigate urban heat or cool islands is the limited number and limited geographical distribution of local weather stations. In addition, not all weather stations produce continuous data. For these reasons, many studies have turned to the use of remotely sensed data in urban heat island analyses [25]. Nasrallah et al. [26] used ground monitoring stations in and around Kuwait City to investigate the presence of urban heat islands.

In this investigation, we made use of available historical satellite data on daytime and nighttime LST to study the spatial distribution of land surface temperature in Kuwait. To the best of our knowledge, there was no previous study that utilized satellite data to characterize temperature differences between urban and non-urban areas in Kuwait. We hypothesized that urban and suburban areas will have higher daytime and nighttime remotely sensed LST compared to rural non-urban areas.

## 2. Materials and Methods

### 2.1. Study Area

Kuwait is located at the northeastern corner of the Arabian Peninsula between 46.5° and 48.5° E and 28.5° and 30.0° N (Figure 1). The landscape slopes gently from about 280 m above sea level in the extreme south-west of the country towards the Arabian Gulf coast in the east [27]. Kuwait shares land borders with Saudi Arabia and Iraq and sea borders with Iran. The total land area is approximately 18,000 km^2^ (nearly the size of New Jersey, USA) with a total population of about 4.12 million in 2017 [28]. It has a large petrochemical industry with associated urban land uses. Kuwait City is the capital and the main city in the country. A land-use survey in 2000 concluded that nearly 75% of the country’s area was rangeland in the form of barren and open sandy fields, while the urbanized area did not exceed 5% of the total land use [29]. The country is divided administratively into six governorates; Al Ahmadi, Al Farwaniyah, Al Jahrah, Al Kuwait (Capital City), Hawalli and Mubarak Al-Kabeer (Figure 1). Based on land cover, governorates that include barren lands were classified as rural governorates. There were two rural governorates: Al Ahmadi and Al Jahrah. We classified Al Farwaniyah, Al Kuwait (Capital City), Hawalli and Mubarak Al-Kabeer as urban/suburban governorates. According to the Kuwait Meteorological Department, seasons in Kuwait are classified as follows: winter (6 Decembe–15 February), spring (16 February–20 May), summer (21 May–4 November) and fall (5 November–5 December) [30].

### 2.2. Data

We used Moderate-Resolution Imaging Spectroradiometer (MODIS) Terra satellite data from the National Aeronautics and Space Administration (NASA). The MOD11A2 (version 6) product from MODIS/Terra includes an 8-day average of land surface temperature and emissivity with a 1 km spatial resolution (each pixel is 1 × 1 km). The user guide explained that the 8-day compositing period was chosen because twice that period is the exact ground track repeat period of the Terra and Aqua platforms [31]. The MODIS/Terra LST products were repeatedly validated over a set of locations and time periods via several ground-truth and validation efforts [32,33,34]. Recent LST products (collection 6 or version 6) have addressed previous accuracy issues, and measurement errors in arid regions for bare soil and validation studies now recommend their application in these regions [35]. The daytime and nighttime LST data for the period from 2001 to 2017 were extracted for Kuwait using Google Earth Engine (https://earthengine.google.com). The data consists of one measure for every 8 days. For better interpretation, we converted the unit of temperature from Kelvin to Celsius. First, we multiplied by a given scale factor (from the user guide) of 0.02 to return the units to Kelvin [31]. Then, we subtracted 273.15 to convert Kelvin to Celsius.

Shapefiles of first-level Kuwait Administrative Divisions (polygons of the six governorates in 2015) were available from the University of California, Berkeley, Museum of Vertebrate Zoology. The shapefiles can be downloaded from the Harvard Geospatial Library open portal (https://hgl.harvard.edu).

### 2.3. Analysis

Over the entire study period and across all pixels, LST was summarized by mean, standard deviation, median, and interquartile range. We plotted LST against time (8-day intervals) for the study period from 2001 to 2017. The descriptive analyses were repeated for both daytime and nighttime LST and were stratified by each governorate. Before visualizing the raster data, we created a normalized land surface temperature (NLST) by scaling the LST values for a given pixel between the minimum (*LST_min_*) and maximum (*LST_max_*) values in each image:
NLST = (*LST* − *LST_min_*)/(*LST_max_* − *LST_min_*)(1)
where the NLST ranges between 0 and 1 and enables the comparison between images calculated from the equation below [36]. An NLST value of 1 represents the maximum temperature and a value of 0 represents the minimum temperature, for a given season or image. We compared images of NLST spatial distribution in winter, spring, summer and fall.

The differences between the average daytime and nighttime LST between each governorate were calculated using analysis of variance (ANOVA) and Tukey’s honest significant difference (HSD) method, a Studentized range statistic. The Tukey’s HSD sets confidence intervals on the differences between the means of the levels of a factor with the specified family-wise probability of coverage. Surface urban heat and cool islands were determined by calculating the average LST of the urban/suburban (T_u/s_) governorate pixels minus the average LST of the rural and barren (T_r_) pixels. A negative difference (T_u/s_ < T_r_) indicates an urban cool effect, while a positive difference (T_u/s_ > T_r_) indicates an urban heat effect.

Extraction, processing and exporting of raster satellite data was done using Google Earth Engine code editor (hands-on tutorials are available from: https://developers.google.com/earth-engine/tutorials). All other analyses were conducted using R software (version 3.6.0) (R Foundation for Statistical Computing, Vienna, Austria).

## 3. Results

Over the entire study period from January 2001 to March 2017, the time trends of 8-day composite LST in Kuwait showed clear seasonal variability (Figure 2). Remotely sensed daytime land surface temperatures were higher than nighttime temperatures, but both showed similar patterns over time. In general, the spatial distribution of NLST in Kuwait showed similar patterns across seasons, but with lesser intensity in fall and winter seasons (Figure 3).

When summarizing the average daytime LST by governorates, we found that urban and suburban governorates had lower temperatures compared to rural governorates, except for Al Farwaniyah. The nighttime LST showed opposite findings, where urbanized areas had higher LST. The mean daytime LST in the capital city (Al Kuwait governorate) was 33.7 ± 3.1 °C, and the nighttime mean was 22.4 ± 0.5 °C. The mean LSTs in Al Jahrah governorate, where most of the large rangeland and open sandy fields are, were 37.2 ± 1.1 °C in the day and 18.3 ± 0.9 °C at night (Table 1).

The results of tests for statistical differences between each governorate are presented in Table 2. The mean daytime LST in the capital city was higher than in Al Jahrah governorate by 3.5 °C (95% CI: 3.1, 3.9, *p* < 0.001), and higher than in Al Ahmadi governorate by 3.2 °C (2.8, 3.6, *p* < 0.001). The nighttime effect was in the opposite direction, Al Jahrah governorate was warmer than the capital city by 4.1 °C (3.8, 4.5, *p* < 0.001), while Al Ahmadi was warmer than the capital city by 3.8 °C (3.4, 4.8, *p* < 0.001). We then combined all pixels of urban and suburban governorates (T_u/s_) and rural governorates (T_r_) and tested for the presence of surface urban heat and cool islands (Table 3). In daytime, the difference of urban and suburban governorates versus rural governorates (T_u/s_ – T_r_) was −1.1°C (−1.2, −1.00, *p* < 0.001) indicating an urban cool effect. At night, the difference (T_u/s_ – T_r_) became 3.6 °C (3.5, 3.7, *p* < 0.001) indicating an urban heat effect.

## 4. Discussion

In this study, we used publicly available historical satellite data to characterize temperature differences between urban and non-urban areas in Kuwait. We found a daytime urban cool island and a nighttime urban heat island. On average, the magnitude of the difference estimate was higher for the nighttime heat effect. This work can inform urban planning policies and help establish spatial temperature exposure data for future population health studies.

To the best of our knowledge, we have identified only one study that investigated urban heat islands in Kuwait. Nasrallah et al. [26] analyzed 23 years (between 1951 and 1980) of maximum and minimum air temperature data from selected monitoring stations in and near Kuwait City, Kuwait. They concluded that there is a general lack of a well-developed heat island in Kuwait. The authors hypothesized that the similarities in the urban and rural landscape of Kuwait City, and its close proximity to a large water body, were the possible explanations for the lack of heat island development. In this investigation, we now show with more spatial granularity a clear development of a daytime urban cool island and a nighttime urban heat island using a different measure of temperature and a different study period. The development of urban heat islands is commonly found in big metropolitan cities in many regions in the world, [25,37,38,39] while the daytime urban cool island seems to be a distinctive feature of arid cities. Our findings of a daytime cool island and nighttime heat island aligned with previous literature from neighboring cities that have similar arid desert climates such as Dubai and Abu Dhabi, United Arab Emirates [40,41], Erbil, Iraq [13], and Tehran, Iran [36].

In general, urban structures and tall buildings absorb heat, reduce air flow, and generate hot air from air conditioner usage [1,2,3,4]. Manmade dark impervious surfaces absorb shortwave radiation and store heat during the daytime and then release longwave radiation slowly at night contributing to the nighttime urban heat island [25,42]. Urban areas also have higher concentrations of cars and other heat-generating activities released from burning fuel [43]. A study by Al-Hemoud et al. [44] identified a surface temperature inversion in urban areas in Kuwait that occurs immediately after sunset and diminishes at daylight after sunrise; this natural phenomenon could add to the occurrence of nighttime surface urban heat islands. On the other hand, daytime urban cool islands can be attributed to several other factors. Soil in urban lands is moist, and daytime evaporation could potentially reduce the surface temperature compared to the dry soil in barren lands in arid areas [13]. Additionally, large water bodies have greater specific heat capacity and provide a potential cooling effect during the daytime [45]. In contrast to the rural areas, the urban areas in Kuwait are located on the Arabian Gulf, which showed significant daytime cool island effect. The temporal variation in LST between coastal and inland pixels can be strongly driven by sea breeze circulation in coastal cities, especially during clear-sky days in hot summers [46]. It is possible that sea breeze is the main driver of lower surface temperatures around the coastal areas in Kuwait. However, we observed daytime urban cool island effects throughout the four seasons and not just in the summer. Additionally, there was an east–west oriented belt of high LST in the inland part of the city (Al Farwaniyah) in the daytime of winter, fall, and faintly in spring (Figure 3). In other words, although a number of arid and desert countries reported an urban cool effect, it seems that the concept is more complicated than is explained by a simple concept of a cool island. More research that can detect high spatial and temporal LST patterns from an aircraft or new-generation geostationary satellite would help urban planners understand urban cool islands better [46].

This study can inform policymakers and urban planners in Kuwait when considering how to relieve the urban heat effect, especially at night. There are several mitigation strategies that have been proposed worldwide to tackle urban warming. Replacing low-albedo surface material (asphalt and concrete) with high-albedo and high-emissivity surfaces may keep the surfaces cooler when exposed to solar radiation [47,48,49]. Vegetation can intercept solar energy, provide shade to surfaces and has higher albedo than pavement; plants absorb and accumulate less heat, while the evapotranspiration process helps to cool the environment [50,51,52]. Green roofs may also contribute to the mitigation of urban heat islands. The albedo value of bitumen, tar, and gravel roofs typically ranges from 0.1 to 0.2, while the albedo value of green roofs is between 0.7 and 0.85 [53]. Experimental evidence showed that large-scale applications of green roofs reduced ambient temperatures by 0.3 to 3 °C [54].

There are a number of limitations to this study. First, we did not compare the satellite data to observed data from ground facilities, as we did not have surface temperature data to reference. Secondly, although satellites provide extensive historical and geographical coverage, we were limited to a 1 km spatial resolution and we could not obtain historical LST data from Google Earth Engine before 2001 to compare our results to the previously published study in 1990 in Kuwait. The spatial resolution of 1 km may not describe finer details of hot or cool spots within the urban city. Additionally, satellites may not be able to accurately capture temperatures from surfaces that are obscured by trees or tall buildings. Data were only available for clear weather conditions and only during the day and night times when the satellite passes over Kuwait. However, we had data approximately every 8 days, which would reduce the amount of missing data. Finally, we did not investigate how land use land cover change influences the intensity of urban heat or cool effects, and we did not investigate spatial differences in LST trends over each year.

## 5. Conclusions

Urbanization in Kuwait has converted natural landscapes of open soil and undisturbed desert areas to manmade engineered surfaces and infrastructure. About 17 years of historical satellite data suggest the presence of a nighttime urban heat island, but also an urban cool island in the daytime. This work can inform climate change adaptation efforts, especially urban planning policies. Our spatial analysis of land surface temperatures in Kuwait can be used to improve temperature exposure assessment for population health studies.

## Figures and Tables

**Figure 1 ijerph-17-02993-f001:**
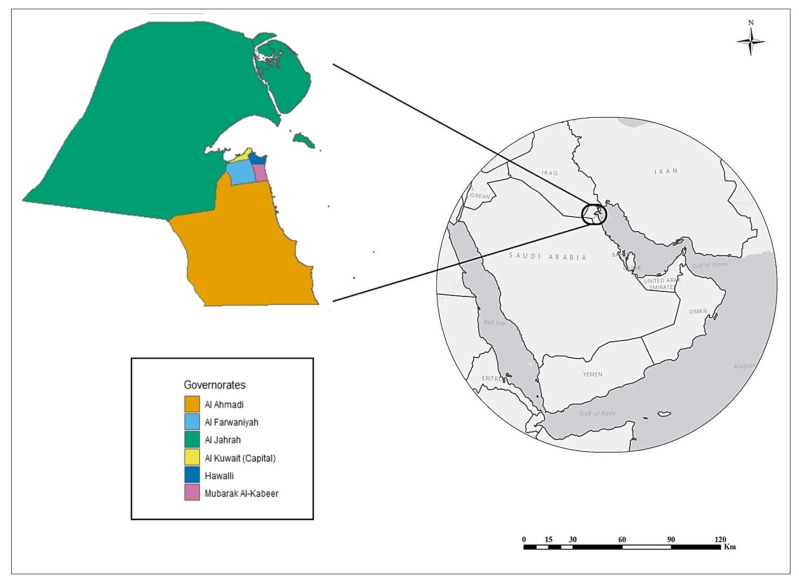
Map of Kuwait and Kuwait’s six governorates.

**Figure 2 ijerph-17-02993-f002:**
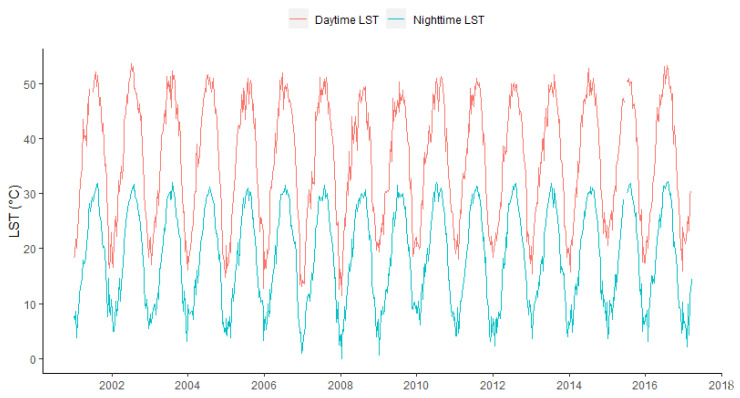
Time trends of land surface temperature (LST) in Kuwait, stratified by daytime and nighttime from January 2001 to March 2017.

**Figure 3 ijerph-17-02993-f003:**
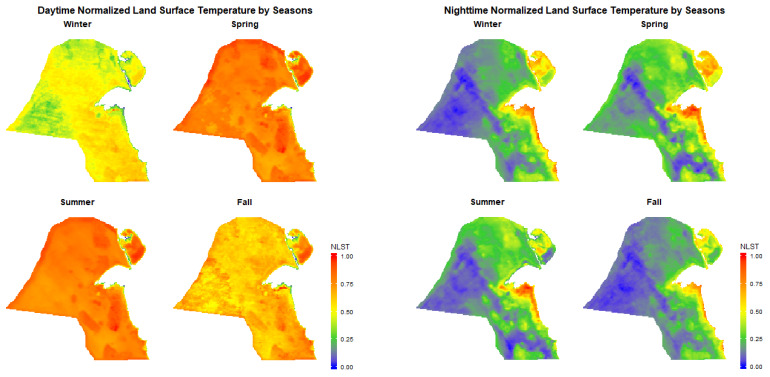
Average daytime and nighttime normalized land surface temperature (NLST) stratified by seasons in Kuwait (2001–2017).

**Table 1 ijerph-17-02993-t001:** Descriptive statistics of average surface temperatures in Kuwait (2001–2017).

Governorate	Mean	SD	Median	IQR	Min	Max
Daytime LST (°C)						
Rural Governorates						
*Al Ahmadi*	36.91	0.89	36.93	1.08	29.9	39.32
*Al Jahrah*	37.24	1.10	37.30	0.90	25.56	40.25
Urban/Suburban Governorates						
*Al Farwaniyah*	37.20	1.30	37.17	1.76	34.92	40.4
*Al Kuwait (Capital City)*	33.74	3.06	33.27	3.63	28.55	40.37
*Hawalli*	35.10	2.74	34.69	4.03	29.12	39.72
*Mubarak Al-Kabeer*	34.86	1.05	34.93	1.02	31.27	36.92
Nighttime LST (°C)						
Rural Governorates						
*Al Ahmadi*	18.57	1.06	18.44	1.36	16.82	23.22
*Al Jahrah*	18.25	0.94	18.15	1.23	16.68	22.25
Urban/Suburban Governorates						
*Al Farwaniyah*	21.52	1.02	21.61	1.72	19.67	23.50
*Al Kuwait (Capital City)*	22.40	0.52	22.38	0.81	21.14	23.38
*Hawalli*	23.05	0.46	23.13	0.58	21.89	23.80
*Mubarak Al-Kabeer*	22.19	0.51	22.35	0.59	20.82	22.98

SD; standard deviation, IQR; interquartile range. LST; land surface temperate over 8 days (in degrees Celsius), Min; average minimum temperature, Max; average maximum temperature.

**Table 2 ijerph-17-02993-t002:** Matrix of differences in land surface temperature between governorates in Kuwait (2001–2017).

	Rural Governorates		Urban/Suburban Governorates			
	Al Ahmadi	Al Jahrah	Al Farwaniyah	Al Kuwait (Captial City)	Hawalli	Mubarak Al−Kabeer
**Absolute Difference in Daytime LST (°C), row minus column (95% CI)**						
***Rural Governorates***						
*Al Ahmadi*		−0.29	−0.32	3.17	1.81	2.06
		(−0.09, −0.49)	(−0.27, −0.37)	(3.59, 2.75)	(2.20, 1.42)	(2.40, 1.71)
*Al Jahrah*	0.32		0.03	3.49	2.13	2.38
	(0.27, 0.37)		(−0.16, 0.23)	(3.91, 3.08)	(2.52, 1.74)	(2.72, 2.04)
***Urban/Suburban Governorates***						
*Al Farwaniyah*	0.29	−0.03		3.46	2.10	2.35
	(0.09, 0.49)	(0.16, −0.23)		(3.92, 3.00)	(2.53, 1.67)	(2.74, 1.96)
*Al Kuwait (Captial City)*	−3.17	−3.49	−3.46		−1.36	−1.12
	(−3.59, −2.75)	(−3.91, −3.08)	(−3.92, −3.00)		(−0.79, −1.93)	(−0.58, −1.65)
*Hawalli*	−1.81	−2.13	−2.10	1.36		0.25
	(−2.20, −1.42)	(−2.52, −1.74)	(−2.53, −1.67)	(0.79, 1.93)		(0.76, −0.27)
*Mubarak Al-Kabeer*	−2.06	−2.38	−2.35	1.12	−0.25	
	(−2.40, −1.71)	(−2.72, −2.04)	(−2.74, −1.96)	(0.58, 1.65)	(−0.76, 0.27)	
**Absolute Difference in Nighttime LST (°C), row minus column (95% CI)**						
***Rural Governorates***						
*Al Ahmadi*		−2.95	0.32	−3.82	−4.48	−3.62
		(−2.77, −3.13)	(0.36, 0.28)	(−3.44, −4.84)	(−4.13, −4.84)	(−3.31, −3.39)
*Al Jahrah*	−0.32		−3.27	−4.14	−4.8	−3.94
	(−0.36, −0.28)		(−3.45, −3.09)	(−3.76, −4.53)	(−4.45, −5.16)	(−3.63, −4.25)
***Urban/Suburban Governorates***						
*Al Farwaniyah*	2.95	3.27		−0.87	−1.53	−0.67
	(2.77, 3.13)	(3.45, 3.09)		(−0.45, −1.29)	(−1.14, −1.93)	(−0.31, −1.02)
*Al Kuwait (Capital City)*	3.82	4.14	0.87		−0.66	0.20
	(3.44, 4.84)	(3.76, 4.53)	(0.45, 1.29)		(−0.14, −1.18)	(0.70, −0.29)
*Hawalli*	4.48	4.8	1.53	0.66		0.86
	(4.13, 4.84)	(4.45, 5.16)	(1.14, 1.93)	(0.14, 1.18)		(1.33, 0.39)
*Mubarak Al-Kabeer*	3.62	3.94	0.67	−0.20	−0.86	
	(3.31, 3.39)	(3.63, 4.25)	(0.31, 1.02)	(−0.70, 0.29)	(−1.33, −0.39)	

LST; land surface temperate (in degrees Celsius). The 95% confidence intervals were constructed from analysis of variance (ANOVA) and Tukey’s honest significant difference (HSD) with specified family-wise probability of coverage.

**Table 3 ijerph-17-02993-t003:** Urban heat and cool effects of urban/suburban and rural governorates in Kuwait.

	Difference	95% CI	*p*-Value	Urban Effect
**Difference in Daytime LST (°C)**				
*T**_Urban_**_/**Suburban**_ – T**_Rural_***	−1.07	−1.17, −0.96	<0.001	Surface urban **cool** island
**Difference in Nighttime LST (°C)**				
*T**_Urban_**_/**Suburban**_ – T**_Rural_***	3.62	3.53, 3.71	<0.001	Surface urban **heat** island

LST; land surface temperate (in degrees Celsius). Urban/Suburban governorates include Al Farwaniyah, Al Kuwait (Capital City), Hawalli and Mubarak Al-Kabeer. Rural governorates include Al Jahrah and Al Ahmadi.

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
