# Peer review of "Spatial Distribution of Land Surface Temperatures in Kuwait: Urban Heat and Cool Islands"

_ijerph, 2020, doi:10.3390/ijerph17092993_

Round 1

Reviewer 1 Report

  • I conclude that this manuscript is suitable for publication AFTER MAJOR REVISION in International Journal of Environmental Research and Public Health. Reviewers' suggestions may be of assistance to you in the preparation of revised papers.

  1. [Citations] All citations are not conform to the instructions for authors of MDPI journal. Please change (No.) to [No.] throughout the manuscript.

  1. [Citations] A blank should be inserted between a word and square bracket [] according to the instructions for authors of MDPI journal. Please check and revise them.

  1. [Title] Land surface temperature (LST) is a popular topic in the academic field of remote sensing. As the authors used “Land surface temperature” in keywords, I suggest the author to change the words “Surface Temperatures” to “Land Surface Temperatures” in the title of this manuscript.

  1. [L124] A fractional equation should be written in formula which follows the “equation format” by instructions for authors of MDPI journal. It is not hard working. Please check and present it in equation format.

  1. [Table 1] Table 1 lists descriptive statistics of LST in Kuwait between 2001 and 2017. In order to compare the daytime/nighttime LST among the governorates, min value and max value should be added in the table.

  1. [Figures 3 and 4] The authors present Figures 3 and 4 to show average daytime and nighttime normalized LST stratified by seasons in Kuwait between 2001 and 2017. However, I think that it is more effective to arrange the figures side by side (seasonal) in order to perform the analysis of heat islands and cold islands due to the difference in LST between day and night in each administrative area of Kuwait. What do the authors think?

  1. [Table 2] In order to show the urban heat and cool effect of urban/suburban and rural governorates, matrix should be ordered in urban, suburban, and rural governorates. However, I found that the Table 2 lists the governorates in alphabetical order. Please explain the reasons for this.

  1. How accurate or reliable in estimating or predicting the land surface temperature using MODIS satellite images in Kuwait? I believe that accuracy or reliability may vary from country to country due to numerous factors and conditions. This is because the effectiveness of the interpretation of the results of this study can only be discussed if this part is presented first.

  1. Do the authors have the LST data across the Kuwait measured or observed by ground facilities? In order to examine the accuracy of LST values derived by MODIS satellite imagery, the estimated LST data should be compared with the measured or observed value (reference values).

  1. [References] All references are not conform to the instructions for authors of MDPI journal. Please check and revise them.

Reviewer 2 Report

The article will be valuable as an analysis of an urban heat island in arid area. However, it is recommended to take the sea breeze effect into consideration in interpreting the result.

The city of Kuwait faces the sea, so that the climate will be influenced by maritime air. It has been found that daytime LST in a coastal city is affected by the sea breeze (e.g., 10.1016/j.uclim.2019.100578). The authors attribute the daytime low LST in the coastal area of Kuwait City to an urban cool island effect, but it is possible that the low LST is affected by the intrusion of a cool sea breeze. Additionally, there is an east-west oriented belt of high LST in the inland part of the city (Al Farwaniyah) in the daytime of winter, fall, and faintly in spring (Fig.3). This is an interesting feature implying that the urban effect on LST distribution is more complicated than is explained by a simple concept of a cool island. It is therefore recommended to make more detailed discussion of the daytime LST distribution.

[Other comments]
@ It is desired to add a map for land use in Kuwait, especially the Kuwait City area, in order to make clear the relationship between the LST distribution and land use.

@ Lines 58-59 "In 2016, the world's highest ambient temperature ever recorded (54.0 C) was seen in Mitribah ---" --- This is the Asian highest, because the world's highest record is regarded to be 56.7C at Death Valley in 1913 (https://wmo.asu.edu/content/world-highest-temperature).

@ There appears to be something wrong with the layout in Line 124.

Round 2

Reviewer 1 Report

The authors have made changes all that I suggested and commented. Revisions have improved the manuscript in all aspects, thus I conclude that this manuscript is suitable for publication .

Reviewer 2 Report

I appreciate the authors' effort of revision. The article is now ready for acceptance.